# Efficacy of Colchicine in Reducing NT-proBNP, Caspase-1, TGF-β, and Galectin-3 Expression and Improving Echocardiography Parameters in Acute Myocardial Infarction: A Multi-Center, Randomized, Placebo-Controlled, Double-Blinded Clinical Trial

**DOI:** 10.3390/jcm14041347

**Published:** 2025-02-18

**Authors:** Tri Astiawati, Mohammad Saifur Rohman, Titin Wihastuti, Hidayat Sujuti, Agustina Tri Endharti, Djanggan Sargowo, Delvac Oceandy, Bayu Lestari, Efta Triastuti, Ricardo Adrian Nugraha

**Affiliations:** 1Doctoral Program of Medical Science, Brawijaya University, Malang 65145, Indonesia; 2Department of Cardiology and Vascular Medicine, Dr. Iskak General Hospital, Tulungagung 66223, Indonesia; 3Department of Cardiology and Cardiovascular Medicine, Faculty of Medicine, Brawijaya University, Dr. Saiful Anwar General Hospital, Malang 65145, Indonesia; ippoenk@ub.ac.id; 4Cardiovascular Research Centre, Universitas Brawijaya, Malang 65145, Indonesia; 5Department of Nursing Science, Faculty of Medicine, Brawijaya University, Malang 65145, Indonesia; titinwihastuti@gmail.com; 6Department of Biochemistry, Faculty of Medicine, Brawijaya University, Malang 65145, Indonesia; hidayatsujuti.fk@ub.ac.id; 7Department of Parasitology, Faculty of Medicine, Brawijaya University, Malang 65145, Indonesia; agustin_almi@ub.ac.id; 8Department of Cardiology and Vascular Medicine, Faculty of Medicine, Brawijaya University, Dr. Saiful Anwar General Hospital, Malang 65145, Indonesia; djanggan@ub.ac.id; 9Division of Cardiovascular Science, Faculty of Biology, Medicine and Health, The University of Manchester, Manchester Academic Health Science Centre, Manchester M13 9PT, UK; delvac.oceandy@manchester.ac.uk; 10Department of Pharmacology, Faculty of Medicine, Brawijaya University, Malang 65145, Indonesia; bayu.lestari@manchester.ac.uk; 11Department of Pharmacy, Faculty of Medicine, Brawijaya University, Malang 65145, Indonesia; efta.triastuti@ub.ac.id; 12Department of Cardiology and Vascular Medicine, Faculty of Medicine, Universitas Airlangga, Dr. Soetomo General Hospital, Surabaya 60286, Indonesia; ricardo.adrian.nugraha-2019@fk.unair.ac.id

**Keywords:** acute myocardial infarction, caspase-1, echocardiography, galectin-3, NOD-like receptor protein 3 inflammasomes, transforming growth factor-β

## Abstract

**Background:** Caspase-1 (reflects NOD-like receptor protein 3 inflammasome activity), transforming growth factor-β (TGF-β), and Galectin-3 play significant roles in post-AMI fibrosis and inflammation. Recently, colchicine was shown to dampen inflammation after AMI; however, its direct benefit remains controversial. **Objectives:** This study aimed to analyze the benefit of colchicine in reducing NT-proBNP, Caspase-1, TGF-β,and Galectin-3 expression and improving systolic–diastolic echocardiography parameters among AMI patients. **Methods:** A double-blinded, placebo-controlled, randomized, multicenter clinical trial was conducted at three hospitals in East Java, Indonesia: Dr. Saiful Anwar Hospital Malang, Dr. Soebandi Hospital Jember, and Dr. Iskak Hospital Tulungagung, between 1 June and 31 December 2023. A total of 161 eligible AMI subjects were randomly allocated 1:1 to colchicine (0.5 mg daily) or standard treatment for 30 days. Caspase-1, TGF-β, and Galectin-3 were tested on day 1 and day 5 by ELISA, while NT-proBNP was tested on days 5 and 30. Transthoracic echocardiography was also performed on day 5 and day 30. **Results:** By day 30, no significant improvements in systolic–diastolic echocardiography parameters had been shown in the colchicine group. However, colchicine reduced the level of NT-proBNP on day 30 more than placebo (ΔNT-proBNP: −73.74 ± 87.53 vs. −75.75 ± 12.44 pg/mL; *p* < 0.001). Moreover, colchicine lowered the level of Caspase-1 expression on day 5 and the levels of TGF-β and Galectin-3 expression on day 1. **Conclusions:** Colchicine can reduce NT-proBNP, Caspase-1, TGF-β, and Galectin-3 expression significantly among AMI patients. Colchicine administration was capable of reducing post-AMI inflammation, ventricular dysfunction, and heart failure but did not improve systolic–diastolic echocardiography parameters (ClinicalTrials.gov identifier: NCT06426537).

## 1. Introduction

After acute myocardial infarction (AMI), the inflammation response is important and necessary for cardiac repair and takes place in the heart to remove the dead tissue resulting from ischemic injury [1]. The infarcted area gives rise to necrotic and stressed cardiomyocytes that trigger inflammation by exposing damage-associated molecular patterns (DAMPs) [2]. However, a persistent long-standing and excessive residual inflammatory response will cause adverse ventricular remodeling [3]. Several major inflammatory mediators in AMI are interleukin-1β (IL-1β) and the NOD-like receptor protein 3 (NLRP3) inflammasome, which play an important role in sterile inflammation triggered by AMI [4].

The NLRP3 inflammasome is considered a very important mediator in the inflammatory reaction after AMI [5,6,7]. Inhibition of the NLRP3 inflammasome is also associated with reduced infarct size in post-ischemic myocardium [8].

Fibrosis of the left ventricular wall is the main change that occurs in ventricular remodeling, and there are several growth factors that play a role in the occurrence of fibrosis after AMI [9]. Transforming growth factor beta (TGF-β) plays an important role in mediating the conversion of fibroblasts into myofibroblasts [10,11,12]. Due to its vital role in myocardial fibrosis and ventricular remodeling, TGF-β has received special attention and has been widely studied in recent years [13]. Several studies have shown that TGF-β expression increases in mouse models of myocardial infarction [14]. As a key signaling molecule, TGF-β deactivates inflammatory macrophages, while promoting myofibroblast trans-differentiation and matrix synthesis through Smad3-dependent pathways in the infarcted heart [14].

Galectin-3 is a protein that plays an important role in the acute phase of tissue repair and ventricular remodeling after AMI [15]. Galectin-3 also plays an important role in the transformation of fibroblasts into myofibroblasts through TGF-β1 pathways [16]. In several studies, galectin-3 has been associated with poor clinical outcomes in heart failure patients [17]. Research on the role of galectin-3 in ventricular remodeling has been conducted in anterior AMI patients receiving percutaneous coronary intervention (PCI) therapy. The study explained that increased galectin-3 levels would increase the risk of left ventricular remodeling [18,19].

The importance of inflammation in myocardial injury and ventricular remodeling after AMI has prompted studies to validate anti-inflammatory agents as potential therapies to prevent ventricular remodeling [20]. Colchicine is one of the anti-inflammatory drugs that is easily available, safe, and inexpensive; it has been widely used in gout arthritis, familial Mediterranean fever, and acute or recurrent pericarditis for decades [21,22]. At lower concentrations (50% inhibitory concentration (IC50) of 3 nM) or typical doses used as prophylaxis, colchicine alters the distribution of E-selectin on endothelial cell surfaces and eliminates the adhesiveness for neutrophils [23]. At higher concentrations (IC50 = 300 nM), colchicine induces shedding of neutrophil adhesion molecules (L-selectin) and prevents further neutrophil recruitment [24]. In a mouse model of non-reperfusion AMI, high-dose colchicine given for 7 days was able to significantly inhibit the increase in NLRP3 and caspase-1 mRNA at 24 h post-AMI [25]. Administration of colchicine to mice undergoing coronary artery ligation and reperfusion can reduce the infarction area and reduce TGF-β levels [26]. In a clinical trial, an initial dose of 1.5 mg colchicine, followed 1 h later by 0.5 mg colchicine daily for 5 days, showed a reduction in myocardial infarct size among patients with STEMI undergoing primary PCI [21]. Another clinical study, the Colchicine Cardiovascular Outcomes Trial (COLCOT), showed a benefit of low-dose colchicine to reduce major adverse cardiovascular outcomes (5.5% vs. 7.1%, *p* = 0.02), with only minor gastrointestinal side effects such as diarrhea and nausea during 22 months of observation [27]. Moreover, our study sought to find a target inflammatory pathway that can be beneficial with these low-dose colchicine.

### Purposes

This study aimed to analyze the efficacy of colchicine in controlling inflammation, ventricular remodeling, and dysfunction after AMI by measuring the level of Caspase-1, TGF-β and Galectin-3 expression on day 5 and measuring NT-proBNP and performing serial transthoracic echocardiography on day 5 and day 30. Our study hypothesis is that the administration of colchicine might reduce the levels of Caspase-1, TGF-β, and Galectin-3 expression on day 5, reduce the level of NT-proBNP on day 30, and improve systolic–diastolic echocardiography parameters on day 30.

## 2. Materials and Methods

### 2.1. Trial Design and Trial Setting

This study was a multi-center, prospective, randomized, double-blind, parallel, placebo-controlled, investigator-initiated, superiority clinical trial. This trial was conducted at 3 sites (3 public hospitals from 3 different cities) in East Java, Indonesia: Dr. Saiful Anwar Hospital in Malang city, Dr. Soebandi Hospital in Jember district, and Dr. Iskak Hospital in Tulungagung district. This study was conducted between 1 June and 31 December 2023. The trial was funded by the authors themselves. The trial protocol was designed by the trial steering committee and registered prospectively at ClinicalTrials.gov (Trial identifier: NCT06426537). The trial protocol was approved by the institutional review board at 3 public hospitals that participated in the trial (Ethical approval number: 400/235/K.3/302/2020). Hard clinical outcomes and potential adverse events were adjudicated by an independent clinical end-point committee composed of senior cardiologists and internists who were unaware of the trial-group assignments. The trial was overseen by a data and safety monitoring board of independent experts. The trial medication and matching placebo were provided by Laboratory of Pharmacology Brawijaya University, which had no role in the design or conduct of the trial or in the preparation or review of the manuscript.

### 2.2. Trial Population

Adult patients aged more than 18 years were eligible to participate if they had had an episode of acute myocardial infarction (AMI) for 24 h, either had any primary PCI procedures or non-reperfusion, and were treated according to national guidelines that included double-antiplatelet and high-dose statin. Diagnostic criteria for AMI were determined according to the Fourth Universal Definition of Myocardial Infarction [28]. Patients were excluded if they had severe cardiogenic shock, history of stroke within the previous 6 months, active malignancy, history of malignancy or autoimmune diseases within the previous 3 years, inflammatory bowel disease, acute liver disease, or liver cirrhosis. Patients with anemia (hemoglobin level < 11.5 g/dL), leucopenia (white blood cell count < 3000 cells/mm^3^), thrombocytopenia (platelet count < 110,000 cells/mm^3^), or severe renal disease (with a serum creatinine level that was greater than two times the upper limit of the normal range) were excluded. Patients that were currently pregnant, breastfeeding, planning to become pregnant during the study, had a prior history of drug or alcohol abuse, on current or planned long-term systemic glucocorticoid therapy, or had a history of hypersensitivity to colchicine were also excluded. Purposive sampling was performed in this study to achieve the minimum number of subjects to participate with the following inclusion and exclusion criteria. Subjects deemed eligible at a screening visit were scheduled for daily visits to the clinic until day 30. We randomly assigned all eligible subjects in a 1:1 ratio to receive either colchicine (at a dose of 0.5 mg once daily) or placebo.

### 2.3. Sample Size Calculation

The sample size in this study was obtained using the formula for clinical superiority design with a continuous variable [29]. The formula description is as follows:$$Sample\;size=2\;\times \;{\left(\frac{z_{1-\alpha }+z_{1-\beta }}{\delta -\delta_{0}}\right)}^{2}\;\times \;s^{2}$$

$z_{1-\alpha }$: The standard normal deviation for colchicine group

$z_{1-\beta }$: The standard normal deviation for standard treatment group

δ: The real difference between two treatment effects

$\delta_{0}$: A clinically acceptable margin

s^2^: Polled standard deviation of both comparison groups

Data regarding mean change in NT-proBNP level were obtained from the previous trial (the COLICA trial), which revealed that the time-averaged reduction in NT-proBNP levels for 4 weeks did not differ between the colchicine group (−57.2%, 95% CI: −64.7% to −48.1%) and the placebo group (−55.3%, 95% CI: −63.0% to −45.7%) [30].

All parameters were assumed as follows: real difference in NT-proBNP between the colchicine group and standard treatment group (δ) = 1.9%; δ_0_ = 1%; α = 0.05; β = 0.20; s = 4 mmHg.$$Sample\;size=2\times {\left(\frac{1.645+0.845}{1.9-1.0}\right)}^{2}\times 4^{2}$$

Then, the minimum sample size requirement was 122.67. To anticipate the problem of loss to follow-up, we decided to add 30% of the sample size; therefore, we recruited a total of 161 subject participants in this study.

### 2.4. Informed Consent

Prior to enrolment, the principal investigator introduced the purpose, benefits, and potential risks of this study to the eligible subjects as well as their families. Subjects had 48 h to consider whether to participate in this clinical trial or not to ensure that participation was entirely voluntary. Moreover, we explained to the subjects that participating or not participating in the study would not affect their standard treatment. Clues to the subjects’ personal information, such as name and hospital number, would be coded instead. The personal information of all participants was always kept confidential.

### 2.5. Randomization and Blinding

Randomization was performed via permuted block randomization through a web-based service (https://www.sealedenvelope.com/randomisation/simulation/), accessed on 4 February 2020. A block size of 4 was considered. Two groups with a 1:1 proportion were formed, where one group received colchicine (at a dose of 0.5 mg once daily) in addition to the conventional therapy provided to both groups. The allocations of treatments are unknown for clinicians, subjects, research assistants, data statistical analysts, and nurses throughout the study (double-blinded).

### 2.6. Trial Intervention

The 161 subjects were randomly assigned in a 1:1 ratio to receive either colchicine (at a dose of 0.5 mg once daily) or placebo. Colchicine (0.5 mg tablet) used in this study was acquired from Lapi Laboratories (Serang, Indonesia). To investigators, the placebo was indistinguishable in packaging, appearance, and sensory properties from the colchicine provided by the pharmaceutical company. A total of 79 subjects in the intervention group received 0.5 mg of colchicine once daily along with the standard treatment for AMI, while a total of 82 subjects in the control group only received the standard treatment for AMI, as can be seen in Figure 1. Concomitant medication was similar in the two groups. Subjects were followed up and scheduled for daily visits to the clinic from day 1 until day 30 of the outpatient treatment phase. Written informed consent was obtained from all the patients before enrollment. Clinical evaluations occurred at 30 days after randomization and every 3 months thereafter. All adverse events and serious adverse events were recorded. The only adverse events recorded were gastrointestinal upset, events that were judged by the data and safety monitoring board of independent experts to be related to colchicine or placebo, or laboratory abnormalities that had been judged by the independent experts to be clinically significant.

### 2.7. Trial Endpoints

The primary efficacy endpoint was change in NT-proBNP level and echocardiography parameters from day 5 to day 30 after AMI. Blood samples were drawn on day 5 and day 30 after AMI. Samples were collected in 161 prechilled tubes containing EDTA, immediately placed on ice, and promptly centrifuged at 4 °C. NT-pro BNP levels were measured with a two-step sandwich ELISA, with 161 streptavidin-coated microtiter plates. Mouse antihuman monoclonal antibody (ABCAM) 184 was used (Elabscience cat. no. E-EL-M0834, Houston, TX, USA). Transthoracic echocardiography was performed on day 30 by Philips AFFINITI I (Koninklijke Philips N.V., Amsterdam, the Netherlands), to measure several echocardiography parameters: ejection fraction (EF) by Teich methods, EF by biplane methods, left ventricular end-systolic volume (LVESV), and left ventricular end-diastolic volume (LVEDV). Tricuspid annular plane systolic excursion (TAPSE) was measured in an apical four-chamber view by placing the 2D cursor at the tricuspid lateral annulus and measuring the distance of systolic annular RV excursion along a longitudinal line defining the end of systole as the end of the T wave in the electrocardiogram. Doppler echocardiography was used to calculate the ratio between early diastolic mitral inflow velocity and early diastolic mitral annular tissue velocity in the septal wall (Septal E/e’), the ratio between early diastolic mitral inflow velocity and early diastolic mitral annular tissue velocity in the lateral wall (Lateral E/e’), the ratio between early (E) and late (A) ventricular filling velocities (E/A ratio), cardiac output (CO), cardiac index (CI), stroke volume (SV) and pulmonary capillary wedge pressure (PCWP) using the Nagueh formula [31]. All echocardiography parameters were measured using standard techniques as described by the European Association of Cardiovascular Imaging (EACVI) [32].

The secondary endpoints consisted of Caspase-1, TGF-β, and Galectin-3 expression on day 1 and day 5 as markers for inflammation. Sera were collected on days 1 and 5, then were centrifuged at 2000–3000 rpm for 20 min. Sera were diluted to 120 μL of standard solution (40 ng/mL) with 120 μL of standard diluent to a standard stock solution of 20 ng/mL. The standard solution was allowed to stand for 15 min with gentle stirring before making dilutions. We prepared duplicate standard points by diluting 1:2 of the standard stock solution (20 ng/mL) sequentially with standard diluent to produce solutions of 10 ng/mL, 5 ng/mL, 2.5 ng/mL, and 1.25 ng/mL. The standard diluent serves as the zero standard (0 ng/mL). The caspase-1 expression in various groups was able to be detected by the caspase-1 activity assay kit (Catalogue number: SAB4503272, Sigma-Aldrich, St. Louis, MO, USA) according to the attached detailed instructions. A total of 40 μL of sample and 10 μL of the rabbit anti-CASP1 (Catalogue number: SAB4503272, Sigma-Aldrich) were added to the sample well, then added with 50 μL of streptavidin-HRP to the sample well and standard well to measure Caspase-1 expression. A total of 40 μL of sample and 10 μL of rabbit anti-TGF-β (Catalogue number: MA1-21595; Thermo Fisher, Waltham, MA, USA) were added to the sample well, then added with 50 μL of streptavidin-HRP to the sample well and standard well (not blank control well). After mixing the well and covering the plate with sealer, we then incubated the sample for 60 min at 37 °C. A total of 40 μL of sample and 10 μL of the mouse anti-Galectin 3 (Catalogue number: MA1-40229; Thermo Fisher) were added to the sample well, then added with 50 μL of streptavidin-HRP to the sample well and standard well (not the blank control well). After mixing the well and covering the plate with sealer, we then incubated the sample for 60 min at 37 °C. We determined the optical density of each well immediately using a microplate reader set to 450 nm within 10 min after adding the stop solution.

### 2.8. Withdrawal Criteria

The withdrawal rates from this trial were shown in Figure 1. Subjects will withdraw from the study for any of the following reasons:Subjects who experience serious adverse events (AEs) throughout the trialSubjects with poor compliance and could not cooperate with clinical examination and follow-upSubjects who quit this clinical trial voluntarily. Subjects can withdraw from the study at any time for any reason. After withdrawal, the subjects’ data will be used with the patient’s consent.

### 2.9. Statistical Analysis

The analyses will adhere to intention-to-treat (ITT) principles. The ITT population included eligible patients who had been randomly assigned to participate in the study, regardless of whether any patients were taking a trial drug. Missing observations are accounted for using the predictive mean matching (PMM) method. Statistical analyses were performed using STATA 14.1 (College Station, TX, USA). Descriptive statistics were used to analyze the results, using mean ± SD, unless stated otherwise. Continuous variables were compared among groups using independent two-tailed *t*-test. The chi-squared test was used to assess the significance of differences between dichotomous variables. Pearson’s or Spearman’s correlation was used to measure correlations between continuous variables. The normality criterion was evaluated using 1-sample Kolmogorov-Smirnov test. The statistical differences were significant if *p*-values < 0.05.

## 3. Results

Of a total of 201 subjects screened with the inclusion and exclusion criteria for the study, 161 subjects were included in the study. From 161 subjects selected to participate in this study, 79 subjects were allocated to the intervention group (colchicine + standard treatment), while 82 subjects were allocated to the control group (standard treatment only). Furthermore, 26 of the randomized patients dropped out, as seen in Figure 1.

### 3.1. Baseline Characteristics of Trial Participants

Demographic and baseline characteristics were similar among both groups (Table 1). No significant differences in the baseline characteristics were found among these groups.

### 3.2. Changes in NT-proBNP During Treatment with Colchicine

As can be seen in Figure 2, mean NT-proBNP on day 5 after AMI was 472.32 ± 90.47 pg/mL in the placebo group (left bar, circle plot) and 418.56 ± 122.63 pg/mL in the colchicine group (left bar, box plot). There was a significant difference between NT-proBNP levels on day 5 after AMI between the colchicine and placebo groups (*p* < 0.001). On day 30 after AMI, mean NT-proBNP was 396.57 ± 42.18 pg/mL in the placebo group (right bar, circle plot) and 144.82 ± 8.51 pg/mL in the colchicine group (right bar, box plot). There was a significant difference between NT-proBNP levels on day 30 after AMI between the colchicine and placebo groups (*p* < 0.001). There was a significant reduction in NT-proBNP levels between day 5 and day 30 (*p* < 0.001); however, mean change in the NT-proBNP level was greater in the colchicine group compared to the control group (ΔNT-proBNP: −273.74 ± 87.53 vs. −75.75 ± 12.44 pg/mL; *p* < 0.001). These changes were consistent when assessing intention-to-treat, as per protocol analysis.

### 3.3. Echocardiography Parameters 30 Days After AMI

All subjects were evaluated with transthoracic echocardiography 30 days after AMI. Transthoracic echocardiography was performed by well-trained cardiologists and sonographers from the three site hospitals. Standard echocardiography parameters were obtained, such as ejection fraction (EF), which is derived from the left ventricular end diastolic volume and left ventricular end systolic volume estimates. EF is measured in a 2D image by Teich methods and a 3D image by the Simpson/biplane method. Tricuspid annular plane systolic excursion (TAPSE) was measured to evaluate global right ventricular function, which describes apex-to-base shortening. Doppler echocardiography was used to calculate several hemodynamic parameters, such as stroke volume (SV), cardiac output (CO), and cardiac index (CI). These parameters are derived from the velocity time integral (VTI) and the cross-section of the left ventricular outflow tract (LVOT). Several parameters were also measured with Doppler Echocardiography, including the ratio between early diastolic mitral inflow velocity and early diastolic mitral annular tissue velocity in the septal wall (Septal E/e’), the ratio between early diastolic mitral inflow velocity and early diastolic mitral annular tissue velocity in the lateral wall (Lateral E/e’), the ratio between early (E) and late (A) ventricular filling velocities (E/A ratio), and the pulmonary capillary wedge pressure (PCWP) using the Nagueh formula [31]. Results can be seen below (Table 2).

### 3.4. Expression of Caspase-1 During Treatment with Colchicine

Figure 3 shows that the expression of Caspase-1, which reflects NLRP3 inflammasome activity after AMI, was not different in the placebo group (left bar, circle plot) compared with the colchicine group (left bar, box plot) 24 h after AMI [4.259 ± 876 vs. 4.871 ± 1276 pg/mL; *p* = 0.384]. On the other hand, the expression of Caspase-1 was significantly higher in the placebo group (right bar, circle plot) compared with the colchicine group (right bar, box plot) 5 days after AMI [4.837 ± 1172 vs. 3.318 ± 976 pg/mL; *p* = 0.013]. Compared with Caspase-1 expression 24 h after AMI, there was a significant reduction in Caspase-1 expression on day 5 in the colchicine group [4.871 ± 1.276 to 3.318 ± 976 pg/mL; Δ = −3.595 ± 1.318 pg/mL; *p* < 0.001] but not in the placebo group [4.259 ± 876 to 4.837 ± 1.172 pg/mL; Δ = +578 ± 118 pg/mL; *p* = 0.235].

### 3.5. Level of TGF-β Expression During Treatment with Colchicine

As can be seen in Figure 4, the level of TGF-β expression was lower in the colchicine group (left bar, box plot) compared with the placebo group (left bar, circle plot) 24 h after AMI [394 ± 276 vs. 863 ± 122 pg/mL; *p* < 0.001]. The level of TGF-β expression was also lower in the colchicine group (right bar, box plot) compared with the placebo group (right bar, circle plot) 5 days after AMI [437 ± 172 vs. 818 ± 116 pg/mL; *p* < 0.001].

### 3.6. Level of Galectin-3 Expression During Treatment with Colchicine

Figure 5 showed that the level of Galectin-3 was lower in the colchicine group (left bar, box plot) compared with the placebo group (left bar, circle plot) 24 h after AMI [417 ± 83 vs. 521 ± 82 pg/mL; *p* = 0.028]. However, the level of Galectin-3 was not different in the colchicine group (right bar, box plot) compared with the placebo group (right bar, circle plot) 5 days after AMI [423 ± 82 vs. 451 ± 86 pg/mL; *p* = 0.451].

## 4. Discussion

Future research directions may also be highlighted. In the last decade, the incidence of AMI has increased among younger patients, and the overall mortality has been increasing. Its morbidity and mortality are the highest among cardiovascular diseases [33]. At present, the main definitive treatments for AMI are thrombolysis and primary PCI, while the long-term treatments include β-blockers, angiotensin-converting enzyme inhibitor (ACE-I), angiotensin receptor antagonists or angiotensin receptor-neprilysin inhibitor (ARNI), high-dose statins, potent antiplatelet drugs, sodium-glucose co-transporter-2 (SGLT-2) inhibitors, and aldosterone antagonists. Although the above treatments improve the prognosis for patients, the mortality of AMI remains high [34]. In the last decade, two classes of anti-diabetic drugs, namely sodium-glucose co-transporter-2 (SGLT-2) inhibitors and glucagon-like peptide-1 (GLP-1) receptor agonists, have attracted scientific interest due to their impressive cardiovascular benefits in AMI patients with and without diabetes. An enormous number of preclinical and clinical studies have addressed their cardiovascular benefits in the setting of AMI [35]. But now, we are trying to move forward to colchicine, a novel agent with anti-inflammatory properties that has attracted notable attention from researchers in various fields because of its powerful pharmacological activities and promising therapeutic prospects in AMI [36]. Colchicine was shown to regulate inflammatory response, oxidative stress, and apoptosis, as well as possess cardioprotective effects against fibrosis and cardiac remodeling [37].

Among patients with stable coronary artery disease, the Low-Dose Colchicine (LoDoCo) trial revealed the benefit of low-dose colchicine [38,39]. In 532 patients with angiographically proven coronary artery disease who were clinically stable for at least 6 months on optimal medical therapy, it was found that colchicine may reduce the risk of the primary composite cardiovascular endpoint of cardiovascular death, MI, ischemic stroke, or ischemia-driven coronary revascularization by 31% compared to placebo (6.8% vs. 9.6%; *p* < 0.001), driven mostly by the occurrence of spontaneous myocardial infarction and ischemia-driven revascularization with median follow-up of 29 months [38,39].

Among patients who had AMI, data regarding the benefit of colchicine is conflicting. The CLEAR trial showed that treatment with colchicine, even started soon after AMI and continued for a median of 3 years, did not reduce the incidence of the composite primary outcome (death from cardiovascular causes, recurrent myocardial infarction, stroke, or unplanned ischemia-driven coronary revascularization) [40]. However, other researchers believe that colchicine may bolster the anti-inflammatory response post-AMI by activating IL-10 pathways in fibroblasts and in clinical settings, potentially reducing inflammation after AMI [41].

Among patients who have had previous coronary artery bypass surgery, the short-term use of colchicine may have a preventive effect that reduces constrictive physiology after 1 month [42]. Among 400 subjects who underwent PCI, the COLCHICINE-PCI trial revealed no differences in major adverse cardiac events but significant decreases in inflammation 24 h after PCI among subjects who received colchicine, as measured by the levels of IL-6 and hs-CRP [43].

These findings are in line with our results, which indicated that the level of inflammation may be reduced with colchicine administration; however, clinical events and echocardiography findings remained similar to those of the control group. There are suggestions that inflammation and natriuretic peptides are linked to one another. People with higher inflammatory markers tend to have higher NT-proBNP levels. Elevated circulating inflammatory markers have been shown to be associated with an increased risk of incident HF, persisting despite adjustment for ‘traditional’ cardiovascular risk factors [44].

The advantage of colchicine remains for myocardial infarction with non-obstructive coronary arteries (MINOCA), which is defined as AMI without any obstructive coronary artery disease (CAD) (≥50% stenosis) in any coronary artery [45]. MINOCA accounts for less than 10% of all patients diagnosed with AMI and carries a 5% risk of mortality at 12 months; the patients are frequently young Asian females with fewer traditional risk factors compared to those with AMI caused by classic CAD [46]. The treatment of MINOCA requires prompt intervention for the myocardial infarction, followed by case-specific management dependent on the underlying cause of the condition. Myocarditis is responsible for a significant portion of cases of MINOCA; however, to our knowledge, there are no data regarding the beneficial effects of colchicine for MINOCA patients [47].

Although anti-inflammatory therapy remains a promising therapeutic option to reduce cardiovascular risk in AMI patients, current findings do not support any benefit of colchicine to reduce MACE, cardiovascular death, or even echocardiography parameters. With regard to all subjects in our study who benefited from low-dose colchicine administration, the benefits proved to be clinically not significant. One reason is because clinical events and echocardiography parameters may require a longer time frame to appreciate, while the level of inflammation can be significantly reduced in just 24 h. Another reason for a lack of benefit with colchicine in our study may be attributable to the pharmacodynamics of colchicine—including too short of a time period for colchicine administration and/or an insufficiently potent dose, particularly in the setting of AMI. Prior reports also support our findings regarding discrepancies between inflammation markers and clinical events after colchicine administration [37,38,39,42].

### Strengths and Limitations

In the current study, we explored the efficacy of colchicine compared to standard treatment among AMI patients. Firstly, a multicenter study was conducted in three different hospitals in East Java with different populations supposedly representative of East Java patients. For quicker recruitment of the necessary number of patients, we selected hospitals that had a huge number of AMI patients in this clinical setting. Secondly, the randomization of subjects to intervention and control groups, with concealed allocation, can minimize selection bias. Thirdly, by keeping both the investigators and subjects blinded to the treatments, this trial helped to control for biases that could have influenced the results. This led to more accurate and reliable results.

The study is subject to several limitations. Firstly, the analyses were limited to the subset of AMI patients with non-reperfusion treatment that in general carried more cardiovascular risk factors compared with real-world evidence. Secondly, due to the short, planned follow-up time, we were unable to understand the impact of long-term results, so we may extend the follow-up time if necessary. Thirdly, we only used single assays to analyze NT-proBNP, Caspase-1, TGF-β, and Galectin-3. Given the substantial assay-specific, glycosylation-dependent, cross-reactivity of the assays, different assays could potentially detect different values. Fourthly, transthoracic echocardiography was performed by different cardiologists from different site hospitals; thus, inter-operator variability cannot be avoided. Fifthly, we did not obtain any echocardiography parameters or NT-proBNP markers upon admission.

## 5. Conclusions

In conclusion, this randomized, placebo-controlled, double-blinded clinical trial innovatively confirmed that colchicine administration may have a potency to reduce post-AMI inflammation by lowering the levels of TGF-β and Galectin-3 expression 24 h after AMI and lowering Caspase-1 expression 5 days after AMI. Colchicine administration was also capable of reducing post-AMI ventricular dysfunction and heart failure by lowering the NT-proBNP level from day 5 until day 30. Unfortunately, colchicine did not improve echocardiography parameters until 30 days of observation. In this study, the benefit of colchicine might appear, at least in part, to reduced the incidence of post-AMI ventricular dysfunction and heart failure linked to inflammatory activity, as well as the elevated risk of heart failure seen in individuals with higher NT-proBNP levels. One important limitation of the study was the relatively small sample size and very short follow-up time. For this reason, the findings cannot be generalized to the broader community based on this study alone.

## 6. Patient and Public Involvement Statement

No subjects or members of the public participated in the conception of our study. However, the results of the study will be published in the appropriate journal after complete data analysis.

## Figures and Tables

**Figure 1 jcm-14-01347-f001:**
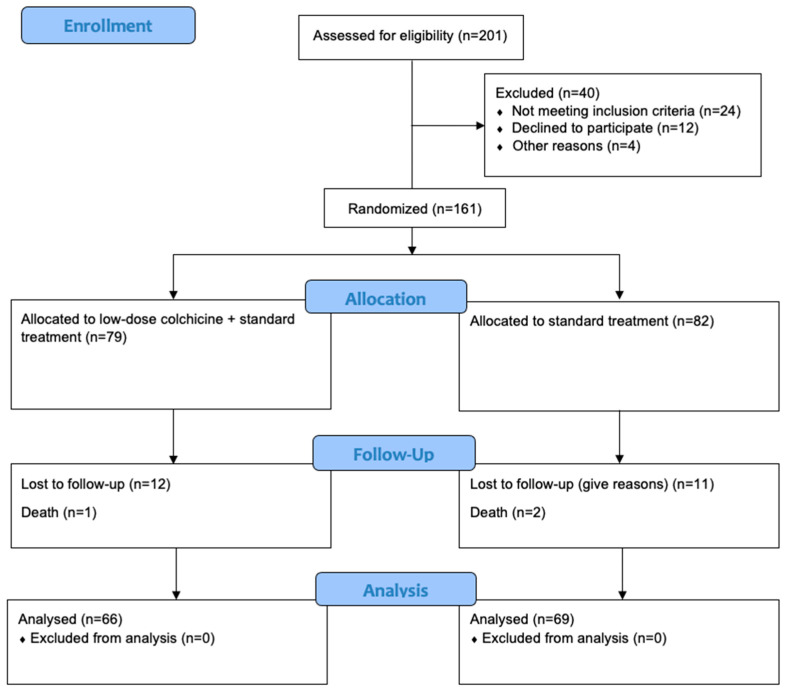
CONSORT flow diagram for study design of this prospective, randomized, double-blind, placebo-controlled, investigator-initiated trial.

**Figure 2 jcm-14-01347-f002:**
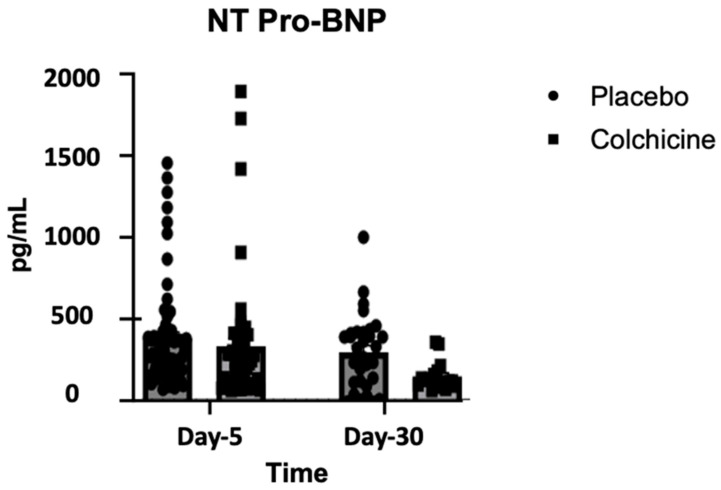
Level of NT-proBNP during experiments, observed on day 5 (left bar) and day 30 (right bar) after AMI. Reduction in NT-proBNP level was significantly greater in the colchicine group compared to the placebo group (*p* < 0.001).

**Figure 3 jcm-14-01347-f003:**
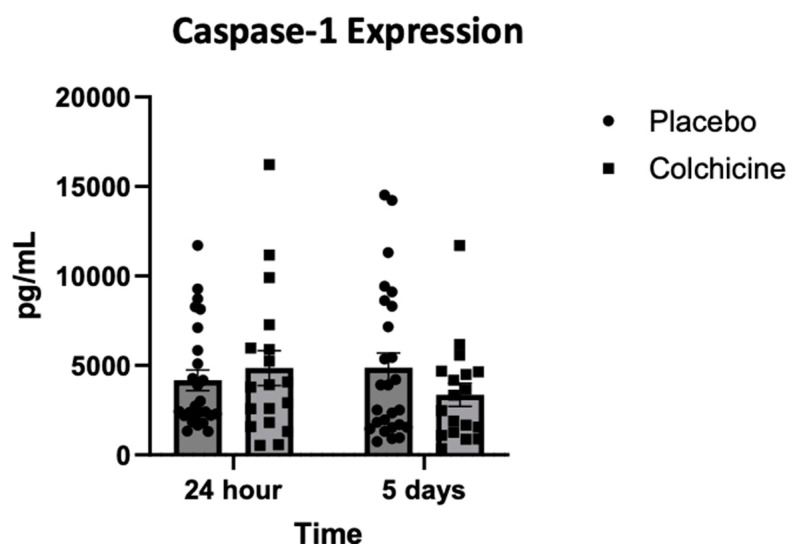
Expression of Caspase-1 (which reflects activity of NLRP3 inflammasome) during experiments, observed 24 h (left bar) and 5 days (right bar) after AMI. Colchicine reduced Caspase-1 expression more on day 5 (*p* < 0.001).

**Figure 4 jcm-14-01347-f004:**
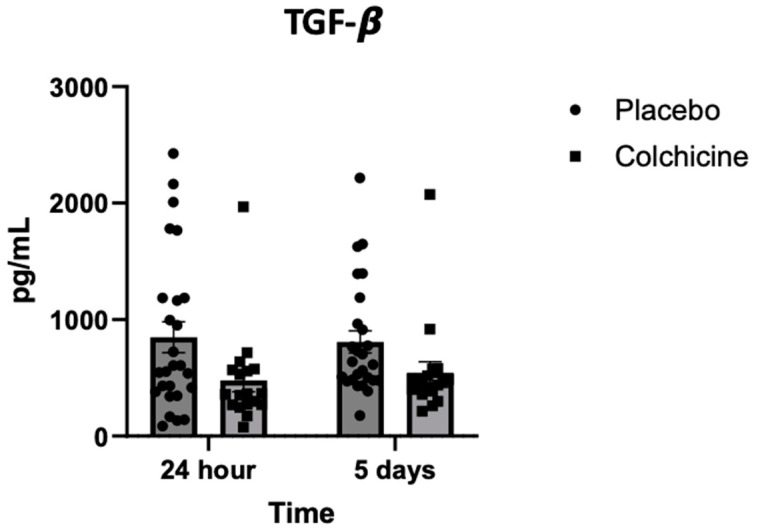
Expression of TGF-β during experiments, observed 24 h (left bar) and 5 days (right bar) after AMI. Compared with placebo, colchicine lowered TGF-β expression 24 h and 5 days after AMI (*p* < 0.001).

**Figure 5 jcm-14-01347-f005:**
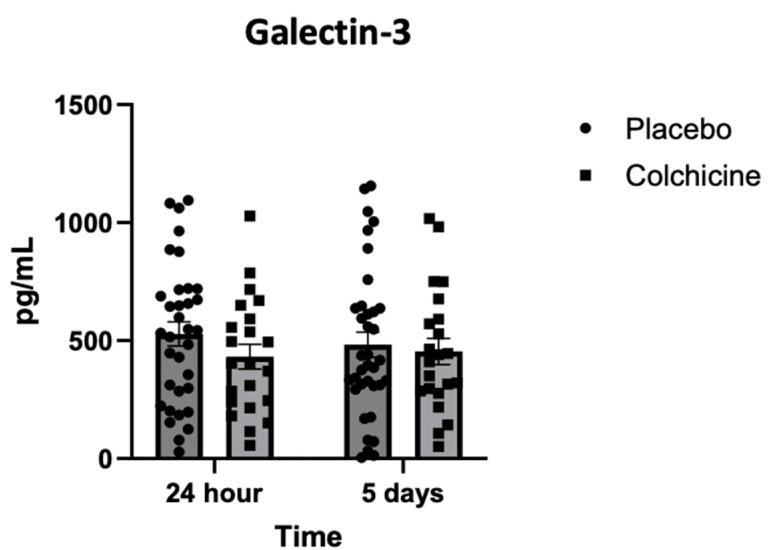
Level of Galectin-3 expression during experiments, observed 24 h (left bar) and 5 days (right bar) after AMI. Compared to placebo, colchicine reduced Galectin-3 expression significantly 24 h (*p* = 0.028), but not 5 days (*p* = 0.451), after AMI.

**Table 1 jcm-14-01347-t001:** Demographic and baseline clinical characteristics of study patients (full analysis set).

Characteristic	Colchicine + Standard Treatment (n = 66)	Standard Treatment Only (n = 69)	*p* Value
Age (mean ± SD, years)	58.27 ± 1.307	55.32 ± 1.45	0.448
Sex (female, %)	33.15 ± 2.44	32.89 ± 2.51	0.136
Body weight (mean ± SD, kilograms)	61.35 ± 1.265	63.30 ± 1.233	0.742
Body height (mean ± SD, centimeters)	161.71 ± 1.026	163.94 ± 0.711	0.324
BMI (mean ± SD, kg/m^2^)	23.43 ± 0.408	23.5 ± 0.35	0.492
Smoker (%)	51.15 ± 8.44	51.27 ± 8.32	0.900
Hypertension (%)	58.66 ± 9.31	58.67 ± 9.29	0.945
Type 2 diabetes (%)	33.56 ± 1.87	33.12 ± 1.69	0.453
Dyslipidemia (%)	72.09 ± 11.88	72.12 ± 12.51	0.895
Systolic blood pressure on ER (mean ± SD, mmHg)	130.32 ± 3.802	132.79 ± 4.136	0.783
Diastolic blood pressure on ER (mean ± SD, mmHg)	80.92 ± 3.12	82.34 ± 2.717	0.584
Heart rate on ER (mean ± SD, bpm)	79.05 ± 3.55	78.08 ± 3.01	0.661
Systolic blood pressure on ICU (mean ± SD, mmHg)	123.24 ± 2.62	124.05 ± 3.49	0.919
Diastolic blood pressure on ICU (mean ± SD, mmHg)	77.32 ± 1.74	78.34 ± 2.66	0.954
Heart rate on ICU (mean ± SD, bpm)	80.97 ±1.88	78.63 ± 2.12	0.993
Killip class (mean ± SD)	1.84 ± 0.52	1.83 ± 0.49	0.899
GRACE score (mean ± SD)	109.27 ± 3.92	104.08 ± 4.95	0.344
TIMI score (mean ± SD)	2.65 ± 0.88	2.67 ± 0.84	0.864
Hemoglobin (mean ± SD, g/dL)	15.30 ± 0.38	14.86 ± 0.46	0.983
Hematocrit (mean ± SD, %)	42.78 ± 1.50	43.50 ± 1.25	0.699
WBC (mean ± SD, cells × 10^9^/L)	14.44 ± 1.15	12.56 ± 0.93	0.992
Platelet count (mean ± SD, cells/mm^3^)	296.20 ± 21.57	284.14 ± 15.79	0.992
Serum creatinine (mean ± SD, mg/dL)	1.33 ± 0.24	1.14 ± 0.07	0.990
eGFR (mean ± SD, mL/min/1.73 m^2^)	71.60 ± 9.46	77.21 ± 5.75	0.675
BUN (mean ± SD, mg/dL)	14.99 ± 1.40	12.38 ± 1.26	0.999
SGOT (mean ± SD, U/L)	111.80 ± 28.15	86.21 ± 16.95	0.252
SGPT (mean ± SD, U/L)	33.70 ± 7.83	37.86 ± 5.76	0.999
Random blood glucose (mean ± SD, mg/dL)	147.64 ± 10.35	143.40 ± 7.49	0.229
HbA1_c_ (mean ± SD, %)	5.97 ± 0.29	5.82 ± 0.13	0.730
Sodium level (mean ± SD, mEq/L)	136.93 ± 0.75	138.50 ± 0.97	1.000
Potassium level (mean ± SD, mEq/L)	4.13 ± 0.11	3.93 ± 0.15	0.996
Chloride level (mean ± SD, mEq/L)	103.43 ± 0.86	103.40 ± 0.69	0.982
Total bilirubin level (mean ± SD, mg/dL)	0.96 ± 0.12	0.96 ± 0.12	0.696
Troponin I (mean ± SD, pg/mL)	7775.89 ± 3640.94	7627.20 ± 4404.32	0.996
Total cholesterol (mean ± SD, mg/dL)	182.86 ± 13.80	176.80 ± 14.26	0.944
LDL (mean ± SD, mg/dL)	138.43 ± 11.97	124.20 ± 13.33	1.000
HDL (mean ± SD, mg/dL)	37.93 ± 3.10	39.70 ± 3.82	0.916
Triglyceride (mean ± SD, mg/dL)	182.29 ± 50.19	164.60 ± 32.95	1.000
Type of acute myocardial infarction			0.987
ST-elevation myocardial infarction (%)	27.56 ± 1.87	27.52 ± 1.69	
Non-ST-elevation myocardial infarction (%)	72.44 ± 3.51	72.48 ± 3.87	

Notes: BMI: body mass index (kg/m^2^); GRACE: The Global Registry of Acute Coronary Events (score 1–372); TIMI: The Thrombolysis in Myocardial Infarction (score: 1–7); WBC: white blood cell count (cells × 10^9^/L); eGFR: estimated glomerular filtration rate (MDRD GFR equation, mL/min/1.73 m^2^); BUN: blood urea nitrogen (mg/dL); SGOT: serum glutamic oxaloacetic transaminase (U/L); SGPT: serum glutamic pyruvic transaminase (U/L); HbA1_c_: hemoglobin A1_c_ (%); LDL: low-density lipoprotein (mg/dL); HDL: high-density lipoprotein (mg/dL).

**Table 2 jcm-14-01347-t002:** Results of transthoracic echocardiography among 135 subjects on day 30 after AMI.

Echocardiography Parameters	Colchicine + Standard Treatment (n = 66)	Standard Treatment Only (n = 69)	*p* Value
EF_Biplane	50.57 ± 3.14	48.23 ± 2.47	0.908
LVESV_Biplane	50.86 ± 4.81	49.90 ± 3.61	0.990
LVEDV_Biplane	97.55 ± 5.79	101.39 ± 5.77	0.815
EF_Teich	52.84 ± 1.95	51.64 ± 2.05	0.878
LVESV_Teich	42.00 ± 2.92	39.68 ± 2.92	0.903
LVEDV_Teich	86.79 ± 4.38	84.71 ± 5.53	0.961
E/E’ Septal	10.45 ± 0.59	11.50 ± 0.53	0.527
E/E’ Lateral	8.92 ± 0.44	9.98 ± 0.87	0.874
EA	0.99 ± 0.07	1,09 ± 0.10	0.897
TAPSE	21.43 ± 0.70	20.55 ± 0.61	0.733
CO	4.29 ± 0.25	4.24 ± 0.29	1.000
CI	2.68 ± 0.18	2.56 ± 0.19	0.966
SV	54.96 ± 3.14	51.28 ± 2.91	0.797
PCWP	13.65 ± 0.56	14.82 ± 0.74	0.572

Notes: EF: ejection fraction (%); LVESV: Left Ventricular End-Systolic Volume (mL); LVEDV: Left Ventricular End-Diastolic Volume (mL); EA: the ratio of the early (E) to late (A) ventricular filling velocities; TAPSE: tricuspid annular plane systolic excursion (mm); CO: cardiac output (L/min); CI: cardiac index (L/min/m^2^); SV: stroke volume (mL/beat); PCWP: pulmonary capillary wedge pressure (mmHg).

## Data Availability

The data that support the findings of this study are available on reasonable request from the corresponding author.

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
