# Peer review of "Efficacy of Colchicine in Reducing NT-proBNP, Caspase-1, TGF-β, and Galectin-3 Expression and Improving Echocardiography Parameters in Acute Myocardial Infarction: A Multi-Center, Randomized, Placebo-Controlled, Double-Blinded Clinical Trial"

_jcm, 2025, doi:10.3390/jcm14041347_

Round 1

Reviewer 1 Report

Comments and Suggestions for Authors

The manuscript aims to evaluate the anti-inflammatory and potential cardioprotective effects of colchicine in post-AMI patients. Overall, the study is well-structured and addresses a critical area of cardiovascular research. The following points should be addressed:

Abstract, The conclusion should explicitly address the clinical implications of colchicine's lack of impact on echocardiography parameters.

Introduction, The discussion of pathways (e.g., NLRP3 inflammasome, TGF-β, Galectin-3) could be summarized to avoid excessive detail.

Clearly distinguish what this study adds compared to prior trials (e.g., COLCOT).

Refocus the purpose statement to emphasize both inflammation markers and clinical outcomes.

Expand the explanation of sample size determination to clarify assumptions used (e.g., power analysis specifics).

Mention corrections for multiple comparisons, if applied.

Echocardiography parameters, While no significant differences were observed, discussing trends (if any) could add depth.

Clarify discrepancies in the reduction of inflammation markers (e.g., Caspase-1, Galectin-3) and clinical outcomes.

Discussion, Strengthen the discussion by comparing results with major trials (e.g., COLCOT, LoDoCo).

Compare and contrast the efficacy of colchicine with other therapies currently under discussion for this population, such as SGLT2 inhibitors and GLP-1 receptor agonists (GLP-1RAs). I would cite the following study DOI: 10.2174/0113816128304097240529053538.

Expand on potential mechanisms explaining the differential impact of colchicine on biomarkers versus clinical outcomes.

Discuss the translational relevance of reduced NT-proBNP and inflammation markers.

Refine the conclusion to balance positive findings (e.g., biomarker reductions) with the absence of echocardiographic improvement.

Some references are outdated (e.g., those prior to 2015). Replace these with more recent reviews or meta-analyses where possible.

Author Response

The manuscript aims to evaluate the anti-inflammatory and potential cardioprotective effects of colchicine in post-AMI patients. Overall, the study is well-structured and addresses a critical area of cardiovascular research. The following points should be addressed:

Abstract, The conclusion should explicitly address the clinical implications of colchicine's lack of impact on echocardiography parameters.

Authors response: Thank you for your suggestion. We have added sentence "Colchicine administration is capable to reduce post-AMI inflammation, ventricular dysfunction and heart failure but didn’t improve echocardiography parameter"

Introduction, The discussion of pathways (e.g., NLRP3 inflammasome, TGF-β, Galectin-3) could be summarized to avoid excessive detail.

Authors response: Thank you for your suggestion. We have deleted several sentences to make the introduction section became simpler.

Clearly distinguish what this study adds compared to prior trials (e.g., COLCOT).

Authors response: Thank you for your suggestion. We have added this statement "Moreover, our study need to find a target inflammatory pathway which can be beneficial with these low dose colchicine."

Refocus the purpose statement to emphasize both inflammation markers and clinical outcomes.

Authors response: Thank you for your suggestion. We have changed our objective statement with "This study aimed to analyze the efficacy of colchicine in reducing inflammation, ventricular remodeling and dysfunction after AMI, by measuring level of Caspase-1, TGF-β and Galectin-3 expression on day 5, measuring NT-proBNP and performing serial transthoracic echocardiography on day 5 and day 30."

Expand the explanation of sample size determination to clarify assumptions used (e.g., power analysis specifics).

Authors response: Thank you for your suggestion. Based on previous study (COLICA trial), we used (δ)=1.9% as the real difference between two treatment effect and δ=1% as a clinically acceptable margin.

Mention corrections for multiple comparisons, if applied.

Authors response: Thank you for your suggestion. We have been adjusting p-values = 0.05 and a 95% of confidence intervals (α=0.05).

Echocardiography parameters, While no significant differences were observed, discussing trends (if any) could add depth.

Authors response: Thank you for your suggestion. Table 2, unfortunately, can only explain transthoracic echocardiography on day-30 after AMI, since trends of regarding these parameters doesn't differ significantly on day-5.

Clarify discrepancies in the reduction of inflammation markers (e.g., Caspase-1, Galectin-3) and clinical outcomes.

Authors response: In the discussion section, with regards to all subjects in our study who got benefit with low-dose colchicine administation, they all proved to be clinically not significant.  One reason is because clinical events and echocardiography parameters may require a longer time frame to appreciate, whilst level of inflammation can be significantly reduced just in 24 hours. Another reason a lack of benefit of colchicine in our study may be attributable to the pharmacodynamics of colchicine—including too short of a time period for colchicine administration and/or an insufficiently potent dosage, particularly in the setting of AMI.

Discussion, Strengthen the discussion by comparing results with major trials (e.g., COLCOT, LoDoCo).

Authors response: Thank you for your suggestion. We have strengthen the findings with others trials such as LoDoCo, LoDoCo2, CLEAR, COLCOT, COLICA and COLCHICINE-PCI Trial.

Compare and contrast the efficacy of colchicine with other therapies currently under discussion for this population, such as SGLT2 inhibitors and GLP-1 receptor agonists (GLP-1RAs). I would cite the following study DOI: 10.2174/0113816128304097240529053538.

Authors response: Thank you for your suggestion. In the paragraph 2 of the discussion section, we have added "In the last decade, two classes of anti-diabetic drugs, namely Sodium-Glucose co-Transporter-2 (SGLT-2) inhibitors and Glucagon-Like Peptide-1 (GLP-1) receptor agonists, have attracted scientific interest due to their impressive cardiovascular beneficial effects in AMI patients with and without diabetes. There is an enormous number of preclinical and clinical studies addressing their cardiovascular benefits in the setting of AMI [43]. But now we try to move forward to Colchicine"

Expand on potential mechanisms explaining the differential impact of colchicine on biomarkers versus clinical outcomes.

Authors response: Thank you for your suggestion. In the last paragprah of the discussion section, we have added "With regards to all subjects in our study who got benefit with low-dose colchicine administation, they all proved to be clinically not significant. One reason is because clinical events and echocardiography parameters may require a longer time frame to appreciate, whilst level of inflammation can be significantly reduced just in 24 hours. Another reason a lack of benefit of colchicine in our study may be attributable to the pharmacodynamics of colchicine—including too short of a time period for colchicine administration and/or an insufficiently potent dosage, particularly in the setting of AMI."

Discuss the translational relevance of reduced NT-proBNP and inflammation markers.

Authors response: Thank you for your suggestion. We have added the sentences : "There are suggestions that inflammation and natriuretic peptides are linked to one another. People with higher inflammatory markers tended to have higher NTproBNP levels. Elevated circulating inflammatory markers have been shown to be associated with an increased risk of incident HF, persisting despite adjustment for ‘traditional’ cardiovascular risk factors [44]."

Refine the conclusion to balance positive findings (e.g., biomarker reductions) with the absence of echocardiographic improvement.

Authors response: Thank you for your suggestion. We have added sentence "In this study, the benefit of the colchicine appears, at least in part, of the incidence of post-AMI ventricular dysfunction and heart failure linked to inflammatory activity, and the elevated risk of heart failure seen in individuals with higher NT-proBNP levels."

Some references are outdated (e.g., those prior to 2015). Replace these with more recent reviews or meta-analyses where possible.

Authors response: Thank you for your suggestion. We try to renew some references.

Reviewer 2 Report

Comments and Suggestions for Authors

The manuscript, entitled "Efficacy of Colchicine in Reducing NT-proBNP, Caspase-1, TGF-β, and Galectin-3 Expression and Improving Echocardiography Parameters in Acute Myocardial Infarction: A Multi-Center, Randomized, Placebo-Controlled, Double-Blinded Clinical Trial," delineates a methodologically sound and clinically pertinent investigation assessing the effectiveness of colchicine in mitigating inflammation and enhancing echocardiographic metrics in patients with acute myocardial infarction (AMI). The randomized, placebo-controlled, double-blind methodology enhances the robustness of the findings. Nonetheless, other aspects necessitate enhancement regarding clarity, methodology, statistical analysis, and the discourse about the study's limits.

The title is verbose and might be more succinct while preserving essential components. The introduction must explicitly articulate the study objectives, accompanied by well-defined hypotheses. The abstract must offer a more accurate quantification of data, and the phrase "echocardiography parameter" lacks specificity; particular parameters should be identified. The methodology section necessitates enhanced specificity regarding the sample size calculation, encompassing the assumptions employed for power analysis, as well as elucidation of the randomization process and allocation concealment techniques. Potential confounders must be addressed, and their control measures delineated, while inclusion and exclusion criteria should be modified to guarantee reproducibility. The primary and secondary outcomes must be clearly delineated.

The selection of statistical tests must be substantiated by references, and the management of missing data, particularly the effect of dropout rates on statistical power, should be discussed. To support claims of statistical significance, it is necessary to present confidence intervals for critical outcomes. The findings section must display baseline characteristics in an organized tabular style, emphasizing intergroup differences, and ensuring data correctness. Echocardiography results must encompass inter-observer variability, and the primary outcome analysis should distinguish between intention-to-treat and per-protocol analyses.

The discourse should be enhanced by elucidating the clinical significance of the findings and clarifying why colchicine failed to enhance echocardiographic parameters despite reductions in biomarkers. The results must be thoroughly compared with the current literature, and the possible mechanisms of action for colchicine beyond its anti-inflammatory effects should be examined. It is essential to note study limitations, including sample size, follow-up time, and potential biases.

The conclusion must acknowledge the study's shortcomings and provide avenues for further research while refraining from overgeneralization. Clear figures, accompanied by legends that elucidate all abbreviations, and well-arranged tables are essential for enhanced readability. Several references are obsolete; it is imperative to incorporate more contemporary works and maintain a uniform citation style throughout the manuscript.

Minor adjustments entail rectifying typographical and grammatical inaccuracies to enhance readability, standardizing nomenclature (e.g., "NT-proBNP" versus "NTproBNP"), and explicitly delineating the study chronology within the methods section.

This study offers significant insights into colchicine's function in AMI management; nevertheless, enhancements in methodological transparency, statistical rigor, and the explanation of therapeutic implications are necessary.

It is advised to rewrite and resubmit with substantial improvements.

Author Response

The manuscript, entitled "Efficacy of Colchicine in Reducing NT-proBNP, Caspase-1, TGF-β, and Galectin-3 Expression and Improving Echocardiography Parameters in Acute Myocardial Infarction: A Multi-Center, Randomized, Placebo-Controlled, Double-Blinded Clinical Trial," delineates a methodologically sound and clinically pertinent investigation assessing the effectiveness of colchicine in mitigating inflammation and enhancing echocardiographic metrics in patients with acute myocardial infarction (AMI). The randomized, placebo-controlled, double-blind methodology enhances the robustness of the findings. Nonetheless, other aspects necessitate enhancement regarding clarity, methodology, statistical analysis, and the discourse about the study's limits.

Authors response: thank you for your suggestion. We have revised the methodology, statistical analysis, and added the study limitation.

The title is verbose and might be more succinct while preserving essential components. The introduction must explicitly articulate the study objectives, accompanied by well-defined hypotheses. The abstract must offer a more accurate quantification of data, and the phrase "echocardiography parameter" lacks specificity; particular parameters should be identified.

Authors response: thank you for your suggestion. We try to shorten the title. We have improved the introduction section with the explicit study objective and well-defined hypothesis. We have revised the phrase "echocardiography parameter" with more specific parameter.

The methodology section necessitates enhanced specificity regarding the sample size calculation, encompassing the assumptions employed for power analysis, as well as elucidation of the randomization process and allocation concealment techniques. Potential confounders must be addressed, and their control measures delineated, while inclusion and exclusion criteria should be modified to guarantee reproducibility. The primary and secondary outcomes must be clearly delineated.

Authors response: thank you for your suggestion. Our assumptions employed for power analysis are based on the previous study (COLICA trial). The primary outcomes are inflammation biomarkers (Caspase-1, TGF-β and Galectin-3 expression) on day 5. The secondary outcomes are level of NT-proBNP on day 30, and systolic-diastolic echocardiography parameter on day 5 and 30.

The selection of statistical tests must be substantiated by references, and the management of missing data, particularly the effect of dropout rates on statistical power, should be discussed. To support claims of statistical significance, it is necessary to present confidence intervals for critical outcomes. The findings section must display baseline characteristics in an organized tabular style, emphasizing intergroup differences, and ensuring data correctness. Echocardiography results must encompass inter-observer variability, and the primary outcome analysis should distinguish between intention-to-treat and per-protocol analyses.

Authors response: thank you for your suggestion. The selection of statistical tests have already be substantiated by references and being discussed with a statistician. Inter-observer variability from echocardiography have already been mitigated by a single professional sonographer who perform the measurements. Our study involved data from the intention-to-treat and per-protocol analyses.

The discourse should be enhanced by elucidating the clinical significance of the findings and clarifying why colchicine failed to enhance echocardiographic parameters despite reductions in biomarkers. The results must be thoroughly compared with the current literature, and the possible mechanisms of action for colchicine beyond its anti-inflammatory effects should be examined. It is essential to note study limitations, including sample size, follow-up time, and potential biases.

Authors response: thank you for your suggestion. In the discussion section, we explained why this discrepancy could be happened. With regards to all subjects in our study who got benefit with low-dose colchicine administation, they all proved to be clinically not significant. One reason is because clinical events and echocardiography parameters may require a longer time frame to appreciate, whilst level of inflammation can be significantly reduced just in 24 hours. Another reason a lack of benefit of colchicine in our study may be attributable to the pharmacodynamics of colchicine—including too short of a time period for colchicine administration and/or an insufficiently potent dosage, particularly in the setting of AMI. Prior reports have also supported our findings about discrepancies between inflammation markers and clinical events after colchicine administration.

The conclusion must acknowledge the study's shortcomings and provide avenues for further research while refraining from overgeneralization. Clear figures, accompanied by legends that elucidate all abbreviations, and well-arranged tables are essential for enhanced readability. Several references are obsolete; it is imperative to incorporate more contemporary works and maintain a uniform citation style throughout the manuscript.

Authors response: thank you for your suggestion. We try to improve our conclusion with some statements to avoid overgeneralization. Several outdated references had already replaced with the newest ones.

Minor adjustments entail rectifying typographical and grammatical inaccuracies to enhance readability, standardizing nomenclature (e.g., "NT-proBNP" versus "NTproBNP"), and explicitly delineating the study chronology within the methods section.

Authors response: thank you for your suggestion. The true nomenclature would be NT-proBNP.

This study offers significant insights into colchicine's function in AMI management; nevertheless, enhancements in methodological transparency, statistical rigor, and the explanation of therapeutic implications are necessary.

Authors response: thank you for your suggestion. We try to improve the methodological transparency, statistical rigor, and the explanation of therapeutic implications. 

It is advised to rewrite and resubmit with substantial improvements.

Authors response: thank you for your suggestion. We have revised the manuscript.

Round 2

Reviewer 1 Report

Comments and Suggestions for Authors

The manuscript has been improved. I have no further comments

Author Response

Thank you so much for your kind words

Reviewer 2 Report

Comments and Suggestions for Authors

No further comments.

Author Response

Thank you so much for your kind words